# The Use of High Initial Doses of Botulinum Toxin Therapy for Cervical Dystonia Is a Risk Factor for Neutralizing Antibody Formation—A Monocentric Cross-Sectional Pilot Study

**DOI:** 10.3390/medicina58010088

**Published:** 2022-01-07

**Authors:** Harald Hefter, Isabelle Schomaecker, Max Schomaecker, Dietmar Rosenthal, Sara Samadzadeh

**Affiliations:** Department of Neurology, Moorenstrasse 5, D-40225 Düsseldorf, Germany; I.Schomaecker@gmail.com (I.S.); MS2368@cornell.edu (M.S.); dietmar.rosenthal@med.uni-duesseldorf.de (D.R.); sara.samadzadeh@yahoo.com (S.S.)

**Keywords:** cervical dystonia, course of disease, botulinum toxin therapy, long-term outcome, secondary treatment failure, antibody formation

## Abstract

*Background and Objectives*: The present study aims to analyze the complex patient/treating physician interaction at onset of botulinum toxin (BoNT) therapy in patients with idiopathic cervical dystonia (CD) and the influence of high initial doses on long-term outcomes. *Materials and Methods*: A total of 74 CD patients with well-documented courses of BoNT treatment were consecutively recruited after written informed consent. Patients had to rate the amount of improvement of CD in percent of severity of CD at onset of BoNT therapy. They had to draw the course of disease severity (CoD) of CD from the onset of symptoms until the onset of BoNT therapy and from the onset of BoNT therapy until recruitment. The remaining severity of CD was estimated by the treating physician using the TSUI score. Demographic- and treatment-related data were extracted from the charts of the patients. Seventeen patients with suspected secondary treatment failure (STF) were tested for the presence of antibodies. *Results*: Depending on the CoD before BoNT therapy, three patient subgroups could be distinguished: rapid onset, continuous onset and delayed onset groups. Time to BoNT therapy, increase in dose and improvement were significantly different between these three groups. In the rapid onset group, with the highest initial doses, the best improvement was reported, but the highest number of patients with an STF and with neutralizing antibodies was also observed. *Conclusion*: The use of high initial doses in the BoNT therapy of CD is associated with a rapid response and quick success; however, it leads to an elevated risk for the development of a secondary treatment failure and induction of neutralizing antibodies.

## 1. Introduction

Cervical dystonia (CD) is the most frequent type of focal dystonia [1,2,3]. CD manifests around midlife, affects about twice as many females compared with males [1] and results from muscular hyperactivity in the shoulder and neck muscles [1,3,4]. Patients may claim a broad spectrum of symptoms, such as pain, muscular hypertension, abnormal position of head and shoulders and tremors and myoclonic jerks [5,6]. These symptoms interfere with everyday life activities and have an impact on social interactions and emotional wellbeing [2,7,8,9]. 

A therapy of choice is a course of intramuscular injections of botulinum neurotoxins (BoNTs), which have to be performed on a regular basis every 3–4 months to achieve a stable plateau of improvement [10,11]. Thus, the immune system of the patients is repeatedly confronted with the traumatic application of a bacterial product, which may lead to the detection of epitopes of the 150 kDa large neurotoxin by dendritic cells, B-cell stimulation and the risk of formation of neutralizing antibodies (NABs) [12]. The development of a partial or even complete secondary treatment failure may be a result [13,14,15,16].

The use of higher doses, duration of treatment and booster injections have been identified as major risk factors for the induction of NABs and the development of PSTF [16,17,18]. Therefore, it has been recommended to treat with BoNT doses as low as possible and as high as necessary [18]. However, “necessary” has not been defined and depends on the aim of treatment. In a larger series of de novo patients, it could have recently been demonstrated that, in patients with an initial low severity of CD, the use of usually high doses did not avoid a further worsening of CD [19]. We therefore have recommended that it is taken into account that CD may progress in patients with a low initial severity; additionally, avoiding low initial doses in these patients is recommended [19]. On the other hand, it has been observed, in patients who developed a complete secondary treatment failure (CSTF) later on, that in contrast to the first injection, the response to the second and following BoNT injections was lower than in patients who did not develop an STF during the following years [20]. It was concluded that NABs are probably induced early in the course of treatment, that it may take a number of injections until the STF becomes clinically overt and to be cautious in the use of high initial doses.

In summary, the initiation of BoNT therapy appears to be a complex interaction between patient and physician concerning optimal long-term outcomes. To analyze this complex interaction in more detail the following study was performed, not only taking into account demographic- and treatment-related data, but also taking into account the entire course of the disease from the onset of symptoms to the onset of BoNT therapy, on the one hand, and from the onset of therapy to the last visit before recruitment on the other hand.

## 2. Methods

The present cross-sectional study was performed according to the declaration of Helsinki and the guidelines for good clinical practice (GCP) and was approved by the local ethics committee of the University of Düsseldorf (number: 4085).

### 2.1. Patients and Treatment-Related Data

Study inclusion criteria were as follows: (i) age over 17; (ii) confirmed diagnosis of idiopathic CD by at least two board-certified neurologists; (iii) written informed consent; (iv) BoNT-naïve patients; (v) start of BoNT therapy at the University hospital of Düsseldorf (Germany); (vi) continuous treatment every 12–13 weeks without any interruption of BoNT therapy of more than 1 treatment cycle; (vii) at least 3 injections of BoNT. Study exclusion criteria were as follows: (i) patient under legal care; (ii) multifocal, segmental and/or symptomatic dystonia at onset of BoNT therapy as documented in the chart; (iii) patients with a pure antecaput or antecollis [21]; (iv) additional disabling disease other than CD; (v) patients with clinical manifestation of disturbances of mood and perception (with special medication or corresponding medical history).

After screening the charts of continuously treated CD patients, 74 patients were consecutively recruited after written consent. Sex, age at recruitment (AGE), age at manifestation of CD (AOS), age at start of BoNT therapy (AOT), duration of BoNT therapy (DURT), initial BoNT preparation, TSUI score at onset of therapy ([22], ITSUI) and initial total dose (IDOSE) were extracted from the charts. The actual severity of CD (ATSUI), the BoNT preparation used and the actual total dose (ADOSE) were documented by the treating physician. To allow comparison of doses of different BoNT preparations, doses were transformed into unified dose units (uDU). Abobotulinumtoxin (aboBoNT/A) doses were divided by 3, rimabotulinumtoxin (rimaBoNT/B) doses were divided by 30 and ona- and incobotulinum-toxin (onaBoNT/A; incoBoNT/A) doses were left unchanged by 3, following evidence-based data and a European consensus paper [3]. The increase of dose (INDOSE) during the course of treatment was calculated as ADOSE-IDOSE. Improvement of the TSUI score (IMPTSUI) during BoNT treatment was calculated as (ITSUI-ATSUI) (*100/ITSUI).

At recruitment, patients assessed the improvement of CD during BoNT therapy in percent of the severity of CD at the onset of therapy (IMPQ). One further outcome measure (IMPD) was determined after the patients drew the temporal development of the severity of CD (course of disease (CoD) graphs; see Section 2.3).

### 2.2. Determination of the Number of Symptoms Per Patient

In the first part of the present pilot study, patients had to recall at the day of recruitment whether the following symptoms had been present at the onset of BoNT therapy: (i) pain in shoulder and/or neck muscles; (ii) elevated muscle tone in shoulder and/or neck muscles; (iii) reduction in head mobility; (iv) abnormal head position; (v) head tremor; (vi) other symptoms specified by the patient. For each item (i)–(v), they had to cross one of two boxes ([yes] or [no]). Thereafter, patients were confronted with this list of symptoms, (i)–(v), and the symptoms the patient had specified before, once again, with the question of whether these symptoms had been present during the last month before recruitment. For each patient, the number of different symptoms (NSPP) was determined for the time before (NSPP-B) and after BoNT therapy (NSPP-A).

### 2.3. Drawing of the Course of Disease Graphs (CoDB and CoDA graphs)

In the second part of this pilot study patients had to produce a graph of the course of disease (CoD) from onset of symptoms to onset of BoNT therapy (CoDB graph) and from onset of BoNT therapy to the day of recruitment (CoDA graph).

For drawing of the CoD graphs a patient was comfortably seated in front of a desk, one hand holding a piece of paper with a square of 10 × 10 cms size and the other holding a pen. To produce a CoDB graph he/she was instructed that he/she had to draw a line without interruptions from the left lower corner of the square (corresponding to 0% severity at onset of symptoms of CD) to the right upper corner of the square (corresponding to 100% severity of CD at onset of BoNT therapy). If the patient did not draw a single line, but produced several line fragments, a second and a third attempt to draw a single line into a new square was allowed, but not a fourth one. The investigator gave verbal support, but not assistance during drawing. No example of a CoDB or CoDA graph was presented.

For drawing of a CoDA graph a second square of 10 × 10 cms size was presented with the left upper corner corresponding to 100% severity of CD at onset of BoNT therapy. On the right edge of the square the patient had to mark the actual severity of CD (ASCD) in relation to severity of CD at onset of therapy. Again the patient had to draw one single line: in case of the CoDA graph from the left upper corner to the ASCD mark. Three attempts were allowed. Improvement of CD estimated by drawing of the CoDA graph (IMPD) was calculated a s (10-ASCD) *10.

After CoD graph drawing patients were clinically examined and thereafter BoNT injected.

Alternative therapies (acupuncture, physiotherapy, etc. [23]) were not controlled in the present study.

### 2.4. Classification of the CoDB and CoDA graphs

74 patients were recruited for CoD graph drawing. Three of them were unable to draw a single line CoDB graph, five of them were unable to draw a single line CoDA graph. Demographic- and treatment-related data of these patients were included in the study (see Table 1). However, their graphs could not be used for further analysis.

A total of 71 CoDB and 66 CoDA graphs were digitized using a commercially available software, DIGITIZEIT^®^, and a standard scanner. After scanning the origin and end of the x-axis, the y-axis and the CoD, the graph had to be specified. When a stick was moved along the scan, the DIGITIZEIT software produced an x–y table, corresponding to the CoD graph, which could be used for further data analysis.

The 71 CoDB graphs were split up into 4 different types of CoDB graphs by visual inspection by 2 independent investigators after a training session with typical examples for each type, as follows: 1—the “rapid onset” (RO) type, which demonstrated a rapid onset followed by a slower progression (Figure 1 (left part)); 2—the “continuous onset” (CO) type, showing a continuous progression from onset of symptoms until initiation of BoNT/A therapy (Figure 1 (middle part)); 3—the “delayed onset” (DO) type, with a slow initial progression but with a faster progression later on (Figure 1 (right part)); finally, 2 “other type” graphs (OO type) which could not be classified as RO, CO or DO type.

The 66 CoDA graphs were split up analogously into 5 different types of CoDA graphs, as follows: 1—the “rapid response” (RR) type, which showed a rapid response and improvement followed by a low plateau of remaining severity of CD; 2—the “continuous response” (CR) type, with a continuous improvement; 3—the “delayed response” (DR) type, with little response or improvement in the beginning; the “STF” types I and II, with a clear improvement in the beginning but a secondary worsening. The patient who had drawn relapses before BoNT therapy (OO type) had also experienced intermittent periods of worsening and relapses after BoNT therapy (OR type). His data were excluded from further analysis. Five examples of typical CoDA graphs are presented in Figure 2.

### 2.5. Classification of the CoDB and CoDA Graphs

After the end of the present study all 17 patients with an STF–CoDA graph type (see Section 2.4) were included in another study on the analysis of neutralizing antibody formation in CD patients with suspected STF (Medical thesis of Beyza Ürer). Blood samples were taken, coded and sent to a blinded contractor (Toxogen^®^, Hannover, Germany). Presence of neutralizing antibodies was analyzed by means of the mouse hemidiaphragm assay (MHDA). All samples were analyzed in one batch. A list of paralysis times of these patients was returned. In patients with a paralysis time >60 mins, the MHDA was classified to be positive.

### 2.6. Statistics

Difference of sex distribution in the four RO, CO, DO and OO patient subgroups were analyzed by means of a chi-squared test. The difference of the AGE, AOS, DURS, DURT, IDOSE, ADOSE, INDOSE, ITSUI, ATSUI, IMPTSUI, IMPQ and IMPD parameters, between the RO, CO and DO subgroups, were tested by means of a three-group ANOVA. The OO subgroup contained only 2 patients and was excluded from the ANOVA analysis. Non-parametric rank correlation coefficients (Spearman’s rho) were used for the correlation of demographic- and treatment-related data. All statistical procedures were part of the SPSS^®^ statistics package (version 25; IBM, Armonk, NY, USA).

## 3. Results

### 3.1. Long-Term Outcomes of the Entire Cohort

The entire cohort was a typical sample of patients with idiopathic CD. The age of onset (AOS) was 45.26 in the mean. Patients were classified according to the cap/col concept [24], determining whether the main component of the abnormal head position was a movement of the head (cap component) or the neck (col component). A proportion of 71% had a torsion of the head on the neck (torticaput), 12% had a tilt of the head on the neck (laterocaput), 7% had a tilt of the neck (laterocollis) and 10% had a pure retrocaput. Patients with a pure antecaput or antecollis component had been excluded (see Methods Section).

The female/male ratio was 1.9 and the time from onset of symptoms to BoNT therapy (DURS) was about 69 months (=5.75 years) with a large variation. Mean initial severity (ITSUI), estimated by means of the TSUI score, was 8.9 with a range between 4 and 13 (for details, see Table 1).

Mean duration of therapy (DURT) was 116 months (=9.66 years) with a large variation. Mean initial dose (IDOSE) was 166 uDU, mean increase of dose (INDOSE) was 50 uDU and the mean TSUI score at recruitment (ATSUI) was 4.4 with a range between 0 and 10 (details in Table 1).

The improvement during therapy was estimated by the treating physicians by the mean of the TSUI score (IMPTSUI), which was 50.6%. When patients were asked to estimate the percentage of improvement of CD during BoNT therapy (IMPQ), they reported 43% in the mean. When they had to mark the improvement on a VAS (IMPD), the improvement was 46% in the mean. For details (mean values and corresponding standard deviations), see Table 1.

Interestingly, IDOSE was negatively correlated with age (r = −0.253; *p* < 0.04), highly correlated with ADOSE (r = 0.677; *p* < 0.009) and highly correlated with ATSUI (r = 0.376; *p* < 0.01). IDOSE was not correlated with ITSUI (r = 0.12; n.s.), but ADOSE was highly significantly correlated with ATSUI (r = 0.497; *p* < 0.009). ATSUI was highly significantly negatively correlated with all three outcome measures (IMPTSUI: r = −0.709, *p* < 0.001; IMPQ: r = −0.437; *p* < 0.009; IMPD: r = −0.340, *p* < 0.01).

Despite a clear improvement (reported by the patients and observed by the treating physicians) and regular treatment every 3–4 months over years, CD significantly progressed. A total of 70 patients completed the questionnaire asking for symptoms before BoNT therapy (Figure 3 left side), and 69 patients completed the questionnaire asking for symptoms during the last month before recruitment (Figure 3 right side). Before BoNT therapy, more than 50% of the patients (*n* = 32) had had only one symptom. After BoNT therapy, the spectrum of the symptoms had become broader: about 2/3 of the patients (*n* = 42) experienced 3 or more symptoms; although, the head position had been improved (see Table 1). The mean number of symptoms was 1.7 before BoNT therapy (NSPP-B) and was 2.7 after BoNT therapy (NSPP-A). Combinations of more than one symptom were significantly (*p* < 0.05) more frequently reported after BoNT therapy. The distributions of symptom combinations were significantly (*p* < 0.05) different before and after BoNT therapy (Figure 3).

### 3.2. Long-Term Outcomes of the Three RO, CO and DO Subgroups

A total of 71 of the 74 CoDB graphs could be used for further analysis. The 71 CoDB graph classifications showed the following distributions: 16 patients (RO group) produced an RO graph, 30 patients (DO group) a DO graph and 23 patients (CO group) a CO graph. The percentage of males was the highest in the RO group. Two patients (OO group) produced graphs with relapses or periods of worsening followed by no change of severity. These two patients were excluded from further analyses.

The mean initial severity of CD (ITSUI) was similar in all three subgroups (see Table 1). However, time to therapy (DURS) was significantly (*p* < 0.01) different. In the rapid onset (RO) group, the age at onset of symptoms was the lowest, as was the mean time to therapy (DURS: <3 years). In the DO group, DURS was longer than 8.5 years.

There was a clear tendency to higher mean initial doses (IDOSE) in the RO group. In this subgroup, only little changes of doses (INDOSE: 7 uDU) were made in contrast to the DO subgroup with a 10-fold higher increase of doses (>73 uDU).

The mean remaining severity of CD after BoNT therapy (ATSUI) was nearly the same in all three subgroups. It was highest in the RO group (4.7) and lowest in the DO group (4.2). However, despite of no significant difference in the improvement of the TSUI score (*p* = 0.37; n.s.) and the worst long-term outcomes (IMPTSUI = 50%), the improvement reported by the patients was the best in the RO group (IMPQ: >56%; IMPD = 66%), in contrast with the patients in the DO group who had the lowest ATSUI and the best long-term outcomes (IMPTSUI = 55.4%), but who reported only 40% improvement in the questionnaire (IMPQ) and marked only 41% improvement on the VAS.

NSPP-B was the same in all 3 subgroups before BoNT therapy (RO group: 1.9, CO group: 1.9, DO group: 1.7) and increased similarly in all 3 patient subgroups during BoNT therapy (NSPP-A: RO group—2.8; CO group—3.0; DO group—2.7).

### 3.3. Development of STF and Antibody Formation in the RO, CO and DO Subgroups

A total 66 of the patients had produced a CoDA graph which could be used for further analysis. The CoDA graphs showed the following distribution: 15 RR graphs (RR group), 21 CR graphs (CR group), 12 DR graphs (DR group) and 1 OR graph (the patient who had drawn relapses before BoNT therapy had also experienced intermittent periods of worsening and relapses after BoNT therapy). A total of 17 of the patients (STFG) produced an STF–CoDA graph, which showed a secondary worsening after an initial good response. Thus, the percentage of patients with STF in our cohort was 17 out of 66 (25.8%). These 17 patients had been tested for the presence of NABs by means of the mouse hemidiaphragm assay (MHDA): 10 (58.8%) were MHDA-positive patients. The incidence of STF (rate of STF divided by DURT) in the entire cohort was 2.67%, and the incidence of NABs (rate of MHDA-positive patients divided by DURT) was at least 1.57%, since not all patients received an MHDA test.

The patients with RR-type CoDA graphs were the “golden responders”. The percentage of patients with RR–CoDA graphs was the highest in the RO group (RO group: 42.9%; CO group: 14.3%; DO group: 20.1%). The distribution of the frequencies of the occurrence of different CoDA graphs did not differ between the RO, CO and DO groups. However, the percentage of secondary treatment failure was the highest in the RO group (RO group: 35.7%; CO group: 28.9%; DO group: 20.1%). The percentage of MHDA-positive patients was also the highest in the RO group (RO group: 35.7%; CO group: 13.0%; DO group: 6.7%).

## 4. Discussion

### 4.1. Initiation of BoNT Therapy Is a Complex Physician/Patient Interaction

The patients in the present study who had presented as BoNT-naïve patients for the first time in our institution to be treated with BoNT had a mean initial severity of CD between 9 and 10 in TSUI score values. This initial severity has already been reported in several studies from our institution [17,18,19,25] and was the same in all three patient subgroups. We therefore think that, regardless of whether the progression of CD is rapid, as in the RO group, or slow, as in the DO subgroup, patients look for help and come to therapy when a certain degree of severity of CD is reached.

Interestingly, the treating physicians in the present retrospective study did not chose the initial dose according to initial severity. There was no correlation between IDOSE and ITSUI (r = 0.12; n.s.). This is in contrast with the highly significant correlation between ADOSE and ATSUI (r = 0.497; *p* < 0.009). During the course of treatment, the treating physicians try to reduce the remaining severity of CD (ATSUI) by increasing the dose to achieve further improvement—producing this highly significant positive correlation between ADOSE and ATSUI results. An increase of dose with duration of therapy has been reported repeatedly [26,27,28].

At onset of BoNT therapy, the treating physicians in the present study had taken into account further aspects in initiating BoNT therapy. The analysis of patient subgroups with a different course of disease before BoNT therapy demonstrated that those patients with a rapid and severe onset of symptoms (RO group) and a short time to therapy (<3 years) were treated with higher doses than patients with the same initial severity but a much longer time to therapy (DO group). This explains why, in the present study, younger patients had been treated with higher initial doses. The treating physicians had taken into account patients’ CD histories and did not treat according to a fixed severity/dose relationship.

### 4.2. Treatment with Initial High Doses: Is This a Good or Bad Treatment Strategy?

Improvements which were experienced and reported by the patients (IMPQ and IMPD) at the day of recruitment were the highest in the RO group, which was treated with the highest initial doses (see Table 1). This was the only group in which patients reported a better outcome than that estimated by the treating physician. This is unusual because treating physicians often overestimate treatment efficacy, since they cannot take into account aspects of CD which they cannot see, such as pain or emotional wellbeing [2,9,28]. Furthermore, these patients contacted our institution and were regularly treated for a long time. Nearly no dose adjustments had to be performed and the percentage of “golden responders” with RR-type CoDA graphs was the highest. Therefore, the use of initial high BoNT doses appears to be a successful management approach for patients with CD.

However, this looks different when immunological aspects come into play. In the RO group, the percentage of patients with a secondary non-response (35.7%), as well as the percentage of NAB-positive patients (35.7%), was the highest. The incidence of STF in the RO group was 2.98% and was higher than in the rest of patients (2.74%). The incidence of NAB induction was 2.98%/year in the RO group, in contrast to 1.14%/year in the rest of the patients. Incidences of NAB induction around 1.0–1.3 have repeatedly been reported [17,18]. An incidence of 2.98 is unusually high. Despite the good clinical outcome, NAB induction was much higher in the RO group, which was treated with higher initial doses.

### 4.3. Implication for Patient Management

The patients in the RO group had a rapidly progressive course of disease of CD before the onset of BoNT therapy. After the onset of BoNT therapy with higher than usual doses (in our institution), more than 40% of them experienced a rapid improvement. This is probably the reason why these patients had a high adherence with the longest duration of therapy and tended to overestimate the improvement, in contrast with the patients in the CO and DO groups.

The long-term outcomes estimated by the treating physicians (ATSUI) were slightly worse in RO group than in the other two subgroups. There was no difference in dose at recruitment (ADOSE) in all three patient subgroups, and progression of CD, estimated by the comparison between NSPP-B and NSPP-A, was the same. However, estimation of improvement by the patients demonstrated that the best experience was made in the RO group.

Unfortunately, the “better feeling” of the RO patients was associated with a more than twofold higher incidence of NAB induction. Therefore, the management of BoNT therapy should not only aim for quick success but should also aim for optimal long-term outcomes after decades. Together with the observation that STF may be induced early in the course of treatment [20], this elicits caution in the application of high doses from the very beginning.

### 4.4. Adaptation of BoNT Treatment to Progression

Cervical dystonia does not remain stable after its first clinical manifestation (see Results Section 3.1 and [6])—it slowly progresses. The dose of BoNT has to be adjusted during the course of disease and treatment. The lack of dose adaptation in the RO group may not necessarily be interpreted as an indicator of good patient management. It may also be interpreted as an indication of a too high initial dose, which was not corrected during the follow-up appointments, with the implication of a high risk of NAB induction and the development of STF.

### 4.5. Limitations of the Study

The present monocentric study heavily relies on the ability of the patients to recall information about their course of disease experienced years or even decades in the past. Disturbances of mood and perception may have influenced the recall. Nevertheless, the majority of patients was able to draw the principal course of disease before and after the onset of BoNT therapy.

Despite these limitations, study data appear to be consistent, allowing the conclusion that BoNT therapy elicits a complex interaction between patient and treating physician, and that the use of initially high doses of BoNT is associated with a higher incidence of NAB induction during long-term BoNT therapy.

To overcome the shortcomings of a monocentric study, a retrospective multi-center study is recommended, comparing the long-term outcomes and the induction of NABs in CD patients who had the same initial severity but were treated with different doses. Ethical aspects aside, a prospective study with a similar design would probably last too long before significant results could be expected.

## Figures and Tables

**Figure 1 medicina-58-00088-f001:**
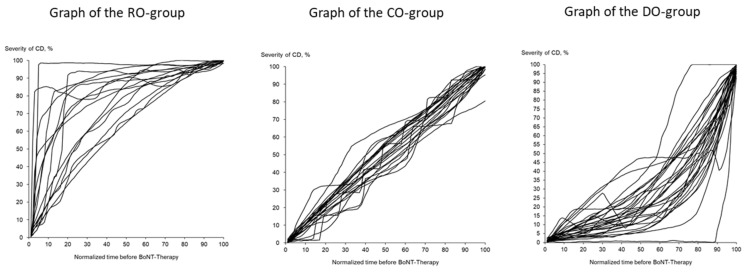
Depending on the curvature, three different main types of course of disease graphs (CoD) could be distinguished. On the left side, all graphs with a rapid onset CoD (*n* = 16) are presented, in the middle, all graphs with a continuous worsening are presented (*n* = 23), and on the right side, all graphs with a low worsening in the beginning (*n* = 30) are presented.

**Figure 2 medicina-58-00088-f002:**
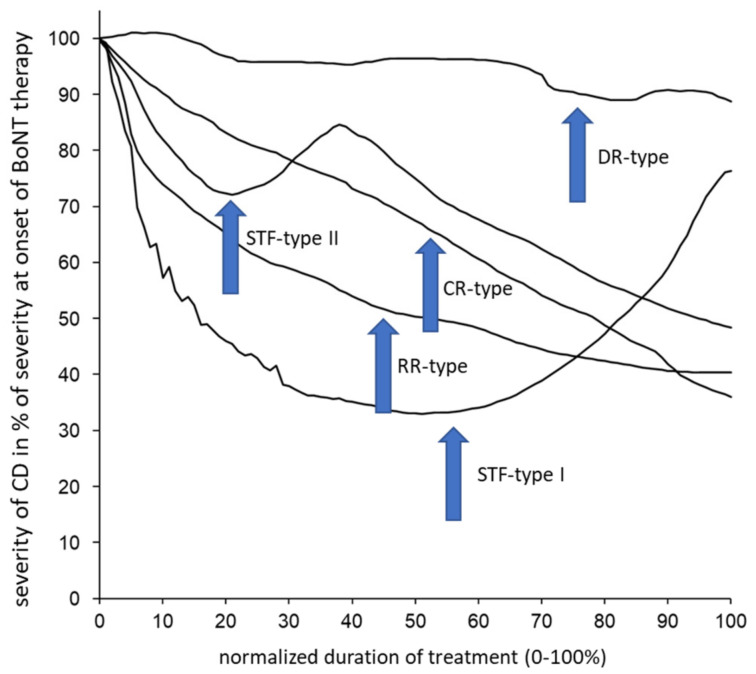
Five different CoDA graph types could be distinguished. One typical example for each CoDA graph type is presented. The RR type goes along with a fast improvement, the CR type has a continuous improvement and the DR type goes along with a slow, small improvement. The STF types may be very different; therefore, two typical examples are presented. The STF type I goes along with a secondary worsening after an initial good response. Switching of the BoNT formulation does not influence the secondary worsening. The STF type II shows an initial response, a secondary worsening and a second response after switch to incoBoNT/A.

**Figure 3 medicina-58-00088-f003:**
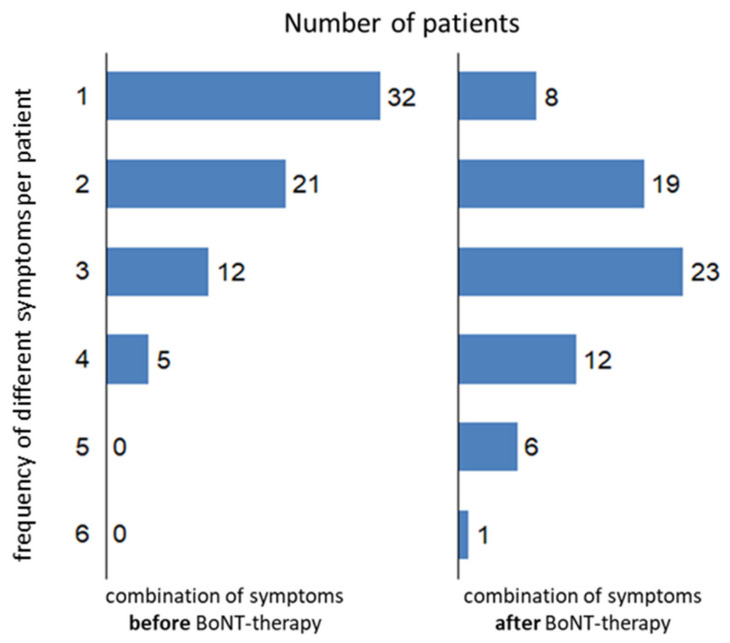
Patients were asked whether a special symptom had been present before the onset of BoNT therapy (left side) and whether it had been present during the last month before recruitment. The number of different symptoms (NSPP) experienced was determined for each patient. The distribution across the cohort of how many patients experienced 1–6 different symptoms is presented in Figure 3. The number of symptoms per patient had a significantly (*p* < 0.05) different distribution before BoNT therapy (left side) compared with the distribution after BoNT therapy (right side). Despite reported improvements, the number of different symptoms per patient increased during BoNT therapy.

**Table 1 medicina-58-00088-t001:** Demographic- and treatment-related data and outcome measures in the three RO, CO, DO subgroups and the entire cohort.

Parameter		RO	CO	DO	ALL	*p*-Value
*n*=		16	23	30	74	
female/male		9/7	19/4	18/12	49/25	0.13; n.s.
AGE	Mean (SD)	57.0 (8.6)	61.0 (13.3)	61.1 (12.1)	60.2 (11.6)	0.49; n.s.
AOS	Mean (SD)	42.9 (11.6)	48.8 (12.5)	43.6 (12.9)	45.26 (12.5)	0.24; n.s.
DURS	Mean (SD)	34.0 (29.3)	46.1 (74.0)	105.2 (114.5)	68.9 (91.5)	*p* < 0.01
DURT	Mean (SD)	144.2 (101.6)	102.3 (72.6)	107.8 (71.1)	115.7 (80.3)	0.25; n.s.
IDOSE	Mean (SD)	205.2 (96.5)	155.2 (62.8)	160.6 (80.1)	166.3 (80.6)	0.14; n.s.
ADOSE	Mean (SD)	212.3 (112.7)	212.8 (108.5)	238.8 (116.2)	217.9 (114.7)	0.65; n.s.
INDOSE	Mean (SD)	7.2 (49.9)	64.8 (77.4)	73.2 (100.1)	50.6 (87.2)	*p* < 0.05
ITSUI	Mean (SD	9.2 (1.8)	8.3 (1.9)	9.2 (3.1)	8.9 (2.4)	0.75; n.s.
ATSUI	Mean (SD)	4.7 (3.1)	4.5 (2.5)	4.2 (2.6)	4.4 (2.6)	0.83; n.s.
IMPTSUI	Mean (SD)	4.6 (3.1)	3.7 (2.7)	5.1 (5.0)	3.9 (3.8)	0.37; n.s.
IMPQ	Mean (SD)	56.7 (30.5)	35.7 (26.4)	40.8 (33.6)	42.9 (31.3)	0.10; n.s.
IMPD	Mean (SD)	66.0 (29.9)	37.0 (25.0)	41.0 (35.0)	46.0 (32.0)	*p* < 0.02

RO—rapid onset subgroup; CO—continuous onset subgroup; DO—delayed onset subgroup (for details see Methods); mean—mean value; SD—standard deviation; AGE(years)—age at recruitment; AOS(years)—age at onset of symptoms; DURS(months)—time from onset of symptoms to BoNT therapy; DURT(months)—duration of BoNT therapy; IDOSE(uDU)—dose at onset of BoNT therapy; ADOSE(uDU)—dose at investigation (actual dose); INDOSE(uDU)—increase of dose during BoNT therapy; ITSUI—TSUI score at onset of BoNT therapy; ATSUI—TSUI at investigation (actual TSUI); IMPTSUI—improvement according to TSUI score; IMPQ—improvement according to questionnaire; IMPD—improvement according to drawing (for details see Methods).

## Data Availability

Data available on request due to restrictions of privacy or ethics. The data presented in this study are available on request from the corresponding author.

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
