# Peer review of "The Use of High Initial Doses of Botulinum Toxin Therapy for Cervical Dystonia Is a Risk Factor for Neutralizing Antibody Formation—A Monocentric Cross-Sectional Pilot Study"

_medicina, 2022, doi:10.3390/medicina58010088_

Round 1

Reviewer 1 Report

1) Some grammatical corrections throughout the text:

*It is advised to read all text searching them

Line 8: this

Line 13: CD (CoD) 1. from

Line 14: and 2. from

Line 64: in clinical practise

Line 72: recommended 1. to take

Line 73: and 2. to avoid

Line 204: Table 1. demographical

2) Please, it is advised to add that the diagnosis of CD was done by at least two board-certified neurologists. Line 93

3) 2.2 Questionnaire to determine the number of symptoms per patient. The questionnaire was validated or is this a pilot study. Please to a description of the questionnaire, include questions, and how was validated. Reference study.

4) 2.3. Drawing of the course of disease graphs. Please same question as to the number 3. Pilot? Validation? Reference study.

5) Please provide baseline characteristics: sex, age, follow-up time, number of BOTOX applications, …

6) Were all the patients followed by the same period?

7) Were all the cervical dystonia of the same type? Please describe how they were (latero, retro, …) in results.

8) Statistics

a) A multivariate analysis is advised to be performed for the exclusion of possible confounding variables.

b) ‘‘A chi-square analysis was performed whether sex distribution was different across these patient subgroups’’ Was only the sex analyzed with chi-square?

c) How did you calculate the power of the study?

d) How was the data about normality?

e) Does Spearman's rank correlation coefficient have a plateau? It is advised to upload as a supplemental file the spearman graph only to the reviewer's purpose for further analysis.

9) Results

  1. a) Instead of / [Table 1 MV/SD 57.0/8.6] use () like [Table 1 MV (SD) 57.0 (8.6)]. This is a standard description.

10) Difficulty understanding the number of patients with NABs. Please provide in the results a clear description of how many patients had been NABs positive.

- ‘‘incidence of NABs was 1.57%.’’

- ‘‘In the RO-group the percentage of patients with a secondary non-response (35.7%) as well as the percentage of NAB-positive patients (35.7%) was the highest.’’

11) Describe in the methods sections that NABs tests were requested.

12) It is advised to update the literature. Only 7/22 of the articles are from the last 5 years of science.

13) The title should mention where the study was done and that it is a retrospective cohort

14) The authors should describe ‘‘What does this manuscript brings new to the present literature?’’. It is well-known that BOTOX doses have an important influence on the development of NABs.

NEW IDEAS:

a) Meta-analysis excluding other confounding variables

b) In the discussion, a figure about the mechanism proposed

c) In the results, a correlation graph

Author Response

Reviewer 1

1) Some grammatical corrections throughout the text:

*It is advised to read all text searching them

Line 8: this

Line 13: CD (CoD) 1. from

Line 14: and 2. from

Line 64: in clinical practise

Line 72: recommended 1. to take

Line 73: and 2. to avoid

Line 204: Table 1. demographical

2) Please, it is advised to add that the diagnosis of CD was done by at least two board-certified neurologists. Line 93

3) 2.2 Questionnaire to determine the number of symptoms per patient. The questionnaire was validated or is this a pilot study. Please to a description of the questionnaire, include questions, and how was validated. Reference study.

4) 2.3. Drawing of the course of disease graphs. Please same question as to the number 3. Pilot? Validation? Reference study.

5) Please provide baseline characteristics: sex, age, follow-up time, number of BOTOX applications, …

6) Were all the patients followed by the same period?

7) Were all the cervical dystonia of the same type? Please describe how they were (latero, retro, …) in results.

8) Statistics

a) A multivariate analysis is advised to be performed for the exclusion of possible confounding variables.

b) ‘‘A chi-square analysis was performed whether sex distribution was different across these patient subgroups’’ Was only the sex analyzed with chi-square?

c) How did you calculate the power of the study?

d) How was the data about normality?

e) Does Spearman's rank correlation coefficient have a plateau? It is advised to upload as a supplemental file the spearman graph only to the reviewer's purpose for further analysis.

9) Results

57 a) Instead of / [Table 1 MV/SD 57.0/8.6] use () like [Table 1 MV (SD) 57.0 (8.6)]. This is a standard description.

10) Difficulty understanding the number of patients with NABs. Please provide in the results a clear description of how many patients had been NABs positive.

- ‘‘incidence of NABs was 1.57%.’’

- ‘‘In the RO-group the percentage of patients with a secondary non-response (35.7%) as well as the percentage of NAB-positive patients (35.7%) was the highest.’’

11) Describe in the methods sections that NABs tests were requested.

12) It is advised to update the literature. Only 7/22 of the articles are from the last 5 years of science.

13) The title should mention where the study was done and that it is a retrospective cohort

14) The authors should describe ‘‘What does this manuscript brings new to the present literature?’’. It is well-known that BOTOX doses have an important influence on the development of NABs.

NEW IDEAS:

a) Meta-analysis excluding other confounding variables

b) In the discussion, a figure about the mechanism proposed

c) In the results, a correlation graph

The first sentence has been modified.

1.       Is omitted.

2.       Is omitted

In clinical practice is omitted.

1.       Is omitted.

2.       Is omitted

This mofified to “Table 1: Demographical

This is indeed the case and has been added.

It is added that this is a pilot study.

There is no reference study.

This is part of the present pilot study.

There is no reference study.

This has been presented in Table 1 for subgroups and the entire cohort.

No. Duration of treatment varied from patient to patient (mean: 115 months; SD: 80.3).

One of the authors (HH) introduced the cap/col-concept. We report on the type of CD in more detail.

Chi2-analysis was only performed sex. This is now mentioned in the statistics section.

For this pilot study we decided to include 75 patients. We did not test a special hypothesis. Therefore, no power calculation was performed.

 We did not test for normality!

We do not know what a spearman graph is. Spearman´s rho is a number used to estimate the correlation of discrete variables.

We are thankful for this advice.

In the revised manuscript we have now added a section on NAB testing.

This sentence is corrected!

This is explained in detail now.

Indeed, we had only a short sentence in line 154/155 on NAB testing. We have added a short separate section on NAB testing.

We have added more recently published articles.

-          We have modified the title.

We now emphasize in more detail that the choice of the initial dose appears to be highly relevant for the long-term outcome and NAB induction.

 In the meantime, we have continued to analyse patient´s drawing in more than 175 patients with CD.

We are thankful to reviewer 1 for his ideas and would include his ideas in a subsequent paper, if reviewer 1 does not mind.

Reviewer 2 Report

The authors present an interesting study on the course of CD in the context of botulinumtoxin injections. Some aspects need to be considered before accepting the manuscript for publication:

-Current standard for statistical reporting should be p=0.xxx throughout the document and figures

-when discussing the limitations and the generalizability/conclusion of your findings, please elaborate on future studies and how they should overcome your shortcomings. 

Author Response

Reviewer 2

The authors present an interesting study on the course of CD in the context of botulinumtoxin injections. Some aspects need to be considered before accepting the manuscript for publication:

-Current standard for statistical reporting should be p=0.xxx throughout the document and figures

-when discussing the limitations and the generalizability/conclusion of your findings, please elaborate on future studies and how they should overcome your shortcomings. 

We are thankful for this comment and have made the necessary corrections.

We have added a short section on future studies.

Round 2

Reviewer 1 Report

1) The title should mention the study type (e.g. cohort) and where was done (e.g. country).

2) Drawing/graphs about the course of the disease. If it was a pilot study the data comparing the graphs and a clearly worsening by days of the disease should be correlated. It is advised to use imaging capture reading packages available for SPSS. This can be uploaded as a supplementary file.

3) Please provide Spearman's rank correlation graphs. They are provided by statistical software. Upload them as supplementary material.

Author Response

Reviewer 1

1)     The title should mention the study type (e.g. cohort) and where was done (e.g. country).

2)     Drawing/graphs about the course of the disease. If it was a pilot study the data comparing the graphs and a clearly worsening by days of the disease should be correlated. It is advised to use imaging capture reading packages available for SPSS. This can be uploaded as a supplementary file.

3) Please provide Spearman's rank correlation graphs. They are provided by statistical software. Upload them as supplementary material.

The title of the revised manuscript is:

The use of high initial doses in the botulinum toxin therapy of cervical dystonia is a risk factor for neutralizing antibody formation - a monocentric cross-sectional pilot study

Reviewer 1 is right: this is a pilot study and the first attempt to analyse the course of disease graphs (CoD-graphs)

Reviewer 1 recommends a very interesting aspect of data processing: to correlate the graphs of the patients with clinical data.

For the CoDB-graphs there are no data available to perform a correlation: the CoDB-graph shows the course of disease from onset of symptoms to onset of BoNT therapy. At the onset of BoNT therapy patients are seen the first time by a neurologist. So there are no data available for correlation.

For the CoDA-graphs the situation is different: for each treatment session a TSUI-score is determined. Therefore, the improvement (not worsening) shown by the graphs produced by the patients can be compared and correlated with the development of TSUI-scores produced by the treating physician. This is work in progress, not trivial to perform and will be presented in a following paper:

For each discrete date of treatment and TSUI-score the graph of the patient has to be interpolated and a corresponding value of the CoDA-graph has to be determined. Then for each patient a correlation between TSUI-scores and CoDA-graph values can be calculated.

Another approach is to interpolate the TSUI-scores so that to each x-value of the CoDA-graph a value of the interpolated TSUI-score can be determined. This will yield similar but not identical results.

Anyway, this is an interesting data analysis strategy which we will complete soon.

We agree with reviewer 1 that it is worth doing, but unfortunately it is not easy to perform. 

We have created two plots:

on the left side the x,y-plot is presented without jitter which shows the exact discrete ATSUI clearly. although covering each other.

On the right side the x,y-plot is presented with jitter so that the individual data can be distinguished.

We hope that is the type of plot reviewer 1 had in mind.